# Effective Microorganisms (EM) Improve Internal Organ Morphology, Intestinal Morphometry and Serum Biochemical Activity in Japanese Quails under *Clostridium perfringens* Challenge

**DOI:** 10.3390/molecules26092786

**Published:** 2021-05-08

**Authors:** Korina Michalska, Michał Gesek, Rajmund Sokół, Daria Murawska, Mateusz Mikiewicz, Agnieszka Chłodowska

**Affiliations:** 1Department of Pathological Anatomy, Faculty of Veterinary Medicine, University of Warmia and Mazury in Olsztyn, Oczapowskiego St. 13, 10-719 Olsztyn, Poland; korina.michalska@gmail.com (K.M.); mateusz.mikiewicz@uwm.edu.pl (M.M.); agnieszka.chlodowska@student.uwm.edu.pl (A.C.); 2Department of Parasitology and Invasive Diseases, Faculty of Veterinary Medicine, University of Warmia and Mazury in Olsztyn, Oczapowskiego St. 13, 10-719 Olsztyn, Poland; rajmund.sokol@uwm.edu.pl; 3Department of Commodity Science and Animal Improvement, Faculty of Animal Bioengineering, University of Warmia and Mazury in Olsztyn, Oczapowskiego St. 5, 10-719 Olsztyn, Poland; daria.murawska@uwm.edu.pl

**Keywords:** effective microorganisms, histopathology, Japanese quail, morphometry, internal organs

## Abstract

The effect of effective microorganisms (EM) on internal organ morphology, intestinal morphometry, and serum biochemical activity in Japanese quails under *Clostridium perfringens* challenge was determined. After 30 days of EM addition, one group of quails was orally inoculated with *Clostridium perfringens*. The second group did not receive EM and was inoculated with *C. perfringens*. In the gut, EM supplementation reduced the number of lesions, enhanced gut health, and protected the mucosa from pathogenic bacteria. EM showed an anti-inflammatory effect and fewer necrotic lesions in villi. In the internal organs, EM showed a protective effect against a typical lesion of *C. perfringens* infection. Necrosis and degeneration of the hepatocytes, necrosis of bile ducts, and bile duct proliferation were more severe in the infected group without EM. Morphometric evaluation showed significantly higher villi in the jejunum after EM addition. A greater crypt depth was observed in the *C. perfringens* group. Biochemical analysis of the blood indicated lower cholesterol on the 12th day of the experiment and between-group differences in total protein, lactate dehydrogenase (LDH), and albumin levels in the EM group. Further studies are needed to improve EM activity against pathologic bacteria as a potential alternative to antibiotics and to develop future natural production systems.

## 1. Introduction

*Clostridium perfringens* (CP) type A is a major agent causing necrotic enteritis in broiler chickens, turkeys, and Japanese quails worldwide, occurring clinically and subclinically, causing loss of performance and financial loss [1,2]. Clostridial organisms, a Gram-positive bacteria, are widely distributed and can be found in the soil, built environment, litter, feed, and intestinal contents of healthy animals and humans. During predisposing factors such as coccidia occurrence, viral infection, and feed abnormalities, damaged intestinal epithelium is a way to colonize *C. perfringens*. Subclinical diseases related to *C. perfringens* pathology occur with decreased digestion, absorption and lower performance [3]. Predisposing factors also result in proliferation of *C. perfringens*, and when concentrations of 10^7^ to 10^9^ are reached, bacterial toxins may induce mucosal damage [4,5].

To control increasing levels of *C. perfringens*, antibiotics are used, but the European Union banned the use of antibiotics as growth promotors; thus, alternatives to antibiotics are needed. In recent years, several experiments have been performed concerning probiotics, prebiotics and synbiotics to find an alternative to antibiotics and to promote animal heath, growth, feed efficiency, performance and intestinal morphology. Timmerman et al. [6] studied seven *Lactobacillus* species on broiler chicken productivity and observed reduced mortality and increased productivity in birds with probiotic supplementation compared to a control group. Kalavathy et al. [7] analysed the effects of 12 strains of *Lactobacillus* on broiler chickens and improved weight gain and feed conversion. Few studies have been conducted evaluating the implication of probiotic organism supplementation during *C. perfringens* experimental infection. The influence of *Bacillus subtilis* improvement under *C. perfringens* was evaluated with the positive effect of probiotic addition improving enterocytes recovery, lower lesion score, and higher intestinal villi [1,8]. Cao et al. [9] challenged *Lactobacillus fermentum* versus *C. perfringens* experimental infection, similar to Cheng el al. [10] with *Bacillus licheniformis* fermented products and Sokale el al. [11] with strains of *Bacillus subtilis* DSM 32315. All mentioned studies showed significantly improved body weight, feed conversion ratio and intestinal lesion score after probiotic supplementation compare to infected chickens.

Additionally, effective microorganisms (EM) are treated as probiotics that promote growth and animal performance. EM are a combination of lactic acid bacteria, photosynthetic bacteria, yeasts, actinomycetes, and fermenting fungi (*Lactobacillus hilgardii*, *L. zeae*, *L. casei*, *L. plantarum*, *L. perolens*, *L. diolivorans*, *Bacillus subtilis, B. thuringiensis*, *B. amyloliquefaciens*, *B. pumilis*, *B. megaterium*, *Acetobacter trophicans*, *A. lovaniensis*, *A. syzygii*, *Saccharomyces cerevisae*, *Moraxella osloensis*, *Rhodopseudomonas palustris*, *Dermacoccus nishinomiyaensis*, *Brachybacterium paraconglomeratum*, *Brevibacillus brevis*, *Devosia chinhatensis*, *Candida ethanothilica*, *Issatchenkia terricola*, *I. orientalis*, *Methylobacterium sorophilicum, Kagangomyces marnhatensis, Candida ethanothilica, Issatchenkia terricola, I. orientosphi,* and *Methylobacterium* spp., measum). A recent paper reported that EM enhance the quality of soil and the growth of plants, suppress odour, prevent infestation with insects, and limit the proliferation of pathogenic bacteria [12,13,14,15].

Many studies include the influence of EM on weight, egg production, egg quality, and the feed conversion ratio in laying hens showing positive effect of EM supplementation on mean weekly egg production and feed conversion [16]. Gnanasesigan et al. [17] conducted similar studies with layers and focused on egg production, mortality, and egg composition and found positive effects after EM addition in feed and water compared to the control group. In broiler chickens, several experiments were conducted, and Jwher et al. [18] reported positive effects on final body weight and morphometry in the jejunum in the EM group. Stęczny and Kokoszyńki [19] examined two EM products in broilers where performance, carcass composition, and microbial contamination were examined (with no significant differences between the EM and control groups) and later EM influenced the morphometric characteristics of the digestive system, leg bones, and caecal microflora with significantly greater total intestinal length, higher intestine-body length ratio, and higher proventriculus percentage in body weight in the EM group [20]. Wondmeneh et al. [21] compared mortality level, weight gain, and feed conversion ratio (FCR) between the control and EM groups of native chickens and found no differences. In Japanese quails, Gesek et al. [22] examined the effect of EM supplementation on intestinal morphology and morphometry (with positive morphometrical data in the EM group), and Sokół et al. [23] evaluated biochemical parameters in quail inoculated with *Cryptospordium parvum* with no significant differences between EM supplemented and EM + *C. parvum* groups.

There are no studies proving the protective effect of EM challenged with *Clostridium perfringens* infection. Our model in this experiment is Japanese quails where enteritis and cholangiohepatitis caused by *Clostridium perfringens* infection were noted with morphological lesions found in the small intestine and liver with similar intensity [24,25]. Thus, the aim of the present study was to determine the effects of EM supplementation during experimental *Clostridium perfringens* infection on the morphology of the internal organs, the morphometry of the intestine (height and width of the villi, depth of the crypt, and muscular layer thickness of the gut in the duodenum, jejunum, and ileum), and serum biochemical parameters in Japanese quails.

## 2. Results

The results of the histopathological evaluation of internal organs are presented in Table 1, Table 2 and Table 3. 

Within the alimentary system (Table 1), infiltrations of lymphoid cells were the main lesions observed in both groups from the duodenum to the caecum. In the duodenum, multifocal infiltration of lymphoid cells was observed more often in the CP group on the 6th day (*P* > 0.05) and 12th day of the experiment (*P* > 0.05), and all CP groups showed a higher number of lesions (*P* > 0.05). In the jejunum on the 3rd day of the experiment, multifocal infiltration of lymphoid cells was more frequent in the CP group (*P* > 0.05), and similar diffuse infiltration of lymphoid cells in the jejunum was also higher on the 3rd (Figure 1A) and 12th days of the experiment (*P* > 0.05), with a higher total number (*P* > 0.05). However, the overall number of focal infiltrating lymphoid cells was higher in the CP + EM group (*P* > 0.05). The necrosis of enterocytes was also more severe in the CP group (*P* > 0.05) (Figure 1). 

Similar to oedema of the villi (*P* > 0.05), fusion of the villi (*P* > 0.05) resulted in increased goblet cell proliferation (*P* > 0.05). Only the irregular surface of the villi was more frequent in the CP + EM group (*P* > 0.05). Within the ileum, the overall number of focal, multifocal, and diffuse infiltration of lymphoid cell lesions was often observed in the CP group (*P* > 0.05). In the liver (Table 2), the CP group showed higher values with significant differences within hydropic degeneration of the hepatocytes (at 12 days and total; *P* > 0.05), fatty degeneration (at 12 days; *P* > 0.05), focal necrosis of the hepatocytes (at 6, 12 days, and total; *P* > 0.05), necrosis of bile ductules (total; *P* > 0.05), and proliferation of bile ductules (total; *P* > 0.05) (Figure 2 and Figure 3). 

In the lungs (Table 2; Figure 4), CP was the group in which morphological lesions were more frequently diagnosed, with infiltration of lymphoid cells in parabronchi/lungs (at 9 days and total; *P* > 0.05). In the heart muscle (Table 3), significant differences were noted for degeneration of cardiomyocytes with vacuolization (total; *P* > 0.05) and atherosclerosis of arteries (at 9 days; *P* > 0.05), diagnosed more often in the CP group. In kidney (Table 3), congestion of glomeruli (12th day; *P* > 0.05), calcium deposits (12th day; *P* > 0.05), and focal necrosis of epithelial cells in tubules (at 12th day and total; *P* > 0.05) occurred more often in the CP group.

Morphometric analysis of diagnosed segments of the alimentary system is presented in Table 4. In the duodenum, significant differences were noted for villus height at 3 days of the experiment (*P* > 0.05) and for thickness of the muscular layer (at 3 and 6 days: *P* > 0.05), with higher values in the CP group. Only at the 12th day, CP + EM showed a thicker wall (*P* > 0.05). Statistical analysis in the duodenum showed that age influenced the villus height (*P* ≤ 0.000; interaction age vs. group *P* = 0.039), crypt depth (*P* = 0.003), and muscular layer thickness (*P* ≤ 0.000; interaction age vs. group *P* ≤ 0.000). In the jejunum, statistical analysis revealed an age influence on the villus width (*P* ≤ 0.000; interaction age vs. group *P* = 0.039) and muscular layer thickness (*P* = 0.009). Additionally, in the jejunum, the influence of villus height (*P* = 0.004) and crypt depth (*P* = 0.043) was noted in the examined group. In the ileum, higher villi were noted at the 12th day of the experiment in the CP + EM group (*P* > 0.05), but earlier, at 3 days, the values were the opposite (*P* > 0.05). When the width of the villi was examined, the CP + EM group showed higher values at day 3 (*P* > 0.05). Crypt depth was greater in the CP group on the 3rd, 6th, and 12th days (*P* > 0.05), and the thickness of the muscular layer was similar on the 3rd day (*P* > 0.05). In the ileum, statistical analysis revealed an age influence on villus height (*P* = 0.030; interaction age vs. group *P* = 0.039), crypt depth (*P* = 0.001), and muscular layer thickness (*P* ≤ 0.000; interaction age vs. group *P* = 0.002). Additionally, in the ileum, the influence of crypt depth (*P* = 0.013) and muscular layer thickness (*P* = 0.001) was noted in the examined group.

Biochemical evaluation (Table 5) revealed higher values with significant differences for cholesterol, total protein, and LDH on the 12th day in the CP group (*P* > 0.05). Statistical analysis of blood data showed that age influenced cholesterol (*P* = 0.004), total protein (*P* ≤ 0.000; the age interaction vs. group *P* ≤ 0.000), and LDH (*P* = 0.003). Between the experimental groups, statistical analysis revealed effects on albumin (*P* = 0.047), total protein (*P* ≤ 0,000), and LDH (*P* ≤ 0.000).

## 3. Discussion

Histopathological analysis of the internal organs showed the most interesting results in the alimentary track and liver. Within the alimentary track, the most pronounced lesion was multifocal infiltration of lymphoid cells in the duodenum, which was observed more often in the CP group than in the CP + EM group. In the jejunum, “focal infiltration” was higher in the CP + EM group, but for diffuse infiltration of lymphoid cells, necrosis of enterocytes, oedema of the villi, and fusion of the villi, the number of lesions was significantly higher in the CP group. Similar in ileum for focal, multifocal, and diffuse infiltration of lymphoid cells. All these lesions are strictly connected with the pathological effect of *Clostridium perfringens* infection. Although we did not observe the characteristic form of clostridial enteritis (necrotic enteritis) during the experiment, our observations were similar to those of Olkowski et al. [26] where subclinical clostridial enteritis was noted. We observed the same morphological lesions, such as strong inflammatory reactions to pathogens within all alimentary tracks, fusion of the villi, oedematous lesions, and hyperaemia. However, similar to Olkowski et al.’s study [26], no typical mucosal necrosis was observed. In the duodenum, we diagnosed a lower number of multifocal infiltrating lymphoid cells in the CP + EM group, and these data are promising compared with Cheng et al. [10]. Those authors did not find lower lesion scores in the duodenum after *Bacillus licheniformis* addition challenged *C. perfringens*; thus, protecting the anti-inflammatory activity caused by EM is visible.

The destructive activity of *C. perfringens* is more pronounced in the jejunum. Inflammatory reaction with diffuse infiltration of lymphoid cells, together with small clusters of necrosis of enterocytes, oedema, and fusion of the villi is the primary stage of necrotic enteritis. A lower number of lesions after EM supplementation enhances jejunal health and protects the mucosa from pathogenic bacteria. A similar protective role was described by Cheng et al. [10] in the jejunum in broilers, who found lower lesion scores after *Bacillus licheniformis* addition after *C. perfringens* experimental infection. Our results also include interesting observations concerning goblet cells in the jejunum. We observed higher goblet cell proliferation in the CP group, in contrast to Abdel-Aziz et al. [27] in Nile tilapia and Reszka et al. [28] in pigs after EM addition. In our opinion, an increased number of goblet cells is a consequence of pathogen stimuli and causes more protective mucus.

In the ileum, similar to the jejunum, histopathological evaluation of CP + EM showed a lower inflammatory reaction to *C. perfringens* infection. The lesser number of lesions after EM addition is similar to Cao et al.’s [9] study where the authors after *Lactobacillus fermentum* addition diagnosed less intense and lower severity of lesions than in the *C. perfringens* group. In contrast to those findings, Cheng et al. [10] did not find a lower lesion score in the ileum after *Bacillus licheniformis* addition was challenged with *C. perfringens*.

In alimentary tracks, the EM addition causes fewer pathological lesions. Aljumaah et al. [8] showed similar results to our data, with lower intestinal lesion scores after *Bacillus subtilis* supplementation than after *C. perfringens* infection. The authors stated that probiotics improve recovery after CP infection. We assert that EM supplementation lowers the number of lesions, enhances gut health, and protects mucosa from pathogenic bacteria.

The second internal organ where morphological lesions were most common was the liver. Degeneration and necrosis of hepatocytes, together with necrosis of bile ductules and bile duct proliferation, are typical changes in *Clostridium perfringens* infection. Lovland and Kaldhusdal [29] observed the same changes in livers during subclinical necrotic enteritis in broilers. The authors noticed bile duct proliferation, necrosis of hepatocytes, and cholangitis, similar to our study. In the same way, Sasaki et al. [30] noted proliferation of bile ductules, fibrosis, and multiple granulomas. Our observations were not severe, as in the above-mentioned studies, but it is worth noting that on the 6th, 9th, and 12th days of the experiment in the CP group, we observed multifocal necrosis of the hepatocytes. Less intense observations were made by Gesek et al. [31] in broiler chickens during the natural course of rearing, and focal necrosis of the hepatocytes, infiltration of lymphoid cells, bile duct proliferation, and cholangiohepatitis were noted, with suspicion of subclinical *C. perfringens* infection.

In our opinion, the lesions observed in the liver are more severe than those observed in the gut. Necrotic lesions and proliferation of bile ductules are more typical for *C. perfringens* infection than inflammatory reactions in alimentary tracks. *Clostridium perfringens* in quails cause more typical lesions for *C. perfringens* in the liver than the alimentary track.

In other internal organs, differences are not as visible. In the heart muscle, degeneration of cardiomyocytes can be correlated with inflammation in the gut and liver. In the kidneys, the necrotic lesions and necrosis of epithelial cells are more pronounced and are more intense in the proximal tubules in the CP group, but focal necrotic lesions are seen in normal laying quails such as those observed in the studies of Gesek et al. [32]. Interestingly, inflammatory lesions and granulomas were observed in the lung. In the absence of bacteriology, it is difficult to connect the infiltration of lymphoid cells and granulomatous inflammation with *Clostridium perfringens* infection because they are more typical of *Escherichia coli* infection. In the case of immunosuppression during enteritis and hepatitis, secondary bacterial co-infection with *Escherichia coli* is possible.

Morphometrical analysis showed some interesting observations. All values concerning villous height, width and crypt depth should be considered together with histopathological lesions observed in the alimentary system. We did not observe typical *Clostridium perfringens* lesions—necrotic enteritis. During necrotic enteritis, necrosis of the villi with haemorrhagic changes was observed, which shortened the villi, and in our study, typical shortening of the villi was not observed. The protective value of probiotic addition was observed in Cheng et al.’s [10] study, where *Bacillus licheniformis* supplementation showed higher villi in the duodenum, jejunum, and ileum compared to the infected *C. perfringens* group. Al-Baadani et al. [33] showed similar observations after probiotic and synbiotic supplementation in the jejunum and ileum. Our data are not promising, and differences within villi height were visible only in the ileum on the 12th day of the experiment. Additionally, in the ileum on the 3rd day, the CP groups showed the narrowest villi, similar to the Aljumaah et al. [8] study with *Bacillus subtilis* and *C. perfringens* challenge and Al-Baadani et al. [33] in the synbiotic group. More typical for clostridial enteritis are our data from crypt depth. Statistical interaction in the jejunum and differences in the ileum at the 3rd, 6th, and 12th day showed deeper crypts during *C. perfringens* infection, related to stimulation of enterocytes to recover, thus the protective role of EM is noted. Sokale et al. [11] noted similar observations in the duodenum after *Bacillus subtilis* addition, and Cheng et al. [10] observed similar observations in the jejunum after *Bacillus licheniformis* supplementation was challenged with *C. perfringens*.

The biochemical analysis of blood showed lower cholesterol levels on the 12th day of the experiment, similar to Wondmeneh et al. [34], where EM in drinking water lowered cholesterol levels compared to control broiler chickens. Abd [35] showed similar data, with lower cholesterol levels in EM-supplemented chickens. Sokół et al. [23] did not find differences in cholesterol levels in EM-supplemented quails. Total protein levels on the 12th day and between experimental groups showed differences, with lower levels in the CP + EM group. These data differ from Sokół et al.’s [23] observations in which no differences were noted. Additionally, LDH levels in the EM-supplemented group showed lower values. Sokół et al. [23] did not find differences in LDH levels in quails. All this information provides promising observations about the protective value of EM, which lowers cholesterol, LDH, and total protein levels in *C. perfringens*-infected quails.

In conclusion, EM supplementation improved the structure and function of the alimentary system as well as the function of internal organs during experimental infection with *Clostridium perfringens*. Protective activity focused mainly on anti-inflammatory effects in the alimentary system, and EM reduced pathological lesions in the liver during *C. perfringens*-related hepatitis. Further studies are needed to evaluate EM activity against other pathologic bacteria (*Escherichia coli* and *Staphylococcus aureus*). Hamad et al. [36] performed microbiological studies with the mentioned bacteria and found an inhibiting role of EM in the growth of pathogenic bacteria. Thus, further studies involving EM in animals are needed, especially in birds, to improve EM as a potential alternative to antibiotics used as growth promoters and to develop future natural production systems.

## 4. Materials and Methods

The study was conducted on 48 14-day-old female Japanese quails (*Coturnix coturnix japonica*). The experiment was approved by the Local Ethics Committee in Olsztyn. The birds were obtained from a commercial farm and were kept in cages in two separate groups of 24 birds each (0.040 m^2^/bird). Examination performed by veterinary poultry specialists did not reveal any signs of bird diseases. No vaccination program was performed and laying quails had 16 h of light. One group of 24 birds (the CP + EM group) received two commercial preparations of EM for 30 days mixed in tap water and in feed from the first day of the experiment. In water, “Multikraft” product in dose 3 L/1000 L of water (Multikraft, Pichl bei Wels, Austria), and according to the manufacturer’s specification contained 1.3 × 10^7^ colony forming units (CFU) of lactic acid bacteria mL^−1^, 3.3 × 10^4^ CFU of photosynthetic bacteria mL^−1^, and 1.3 × 10^4^ CFU yeasts mL^−1^—as previously described [22].

The “MultiPROFIT” product was used in feed at a dose of 5 kg/1000 kg mixed in a standard diet, and according to the manufacturer’s specification, it contained 1.3 × 10^7^ CFU of lactic acid bacteria mL^−1^, 3.3 × 10^4^ CFU of photosynthetic bacteria mL^−1^, and 1.3 × 10^4^ CFU yeasts mL^−1^ (MultiPROFIT, Agro Concept, Puławy, Poland)—as previously described [22]. Both products were mixed with water and feed every day before supplementation. The animals were fed ad libitum, and mixed products in feed and water were available all day. The second group of 24 birds received an ad libitum standard diet and fresh drinking water (without EM products), and Table 6 presents the feed parameters. After 30 days, all birds (44-day-old layers) were infected with a dose of *Clostridium perfringens*. A cocktail of the *C. perfringens* type A (approximately 1 × 10^9^ CFU/mL) strain recovered from severe intestinal lesions in commercial broilers, turkeys, and quails with enteritis. Fresh broth culture was individually administered to each bird by oral gavage (1.0 mL/ bird). Every 3 days (3, 6, 9, and 12 days after infection), blood samples from the brachial/wing vein from six birds from both groups were collected into tubes containing ethylenediaminetetraacetic acid (EDTA) and centrifuged at 2500× *g* for 10 min. Then, quails were slaughtered by cervical dislocation. Aspartate aminotransferase (AST) and alanine aminotransferase (ALT) serum activity were determined by the International Federation of Clinical Chemistry (IFCC) kinetic method, LDH serum activity was determined by the Deutsche Gesellschaft für Klinische Chemie (DGKC) kinetic method, total protein (TP) levels were determined by the biuret test, albumin levels were determined by the bromocresol green method, and cholesterol levels were determined by the colorimetric method with cholesterol esterase and oxidase. Measurements were performed with the use of the Cormay^®^ACCENT-200 chemistry analyser and Cormay^®^ reagents. Two-way ANOVA was performed (the effects of age and EM addition were analysed). The statistical analysis included the determination of the characteristics of the analysed traits (the arithmetic mean and SEM) and the significance of differences in the mean value between age and EM addition groups by Duncan’s D test (significance was set at *P* ≤ 0.05). The research hypothesis was verified with regard to histopathological changes by the chi-squared test (significant at *P* ≤ 0.05). Computations were performed using STATISTICA 10.0 software (StatSoft Inc. 2017).

During necropsy, samples of internal organs were taken for histopathological and morphometric examination. From alimentary system samples of the duodenum (medial portion), jejunum (medial portion posterior to the bile ducts and anterior to Meckel’s diverticulum), ileum (medial portion posterior to Meckel’s diverticulum and anterior to the ileocecal junction), and caecum (medial portion) were collected. Then, heart, lung, kidney, and liver samples were also collected. All tissues were fixed in 10% neutralized formalin and embedded in paraffin blocks for microscopic evaluation. The paraffin sections (4 µm) were stained with haematoxylin and eosin. Microscopic subjective evaluation was conducted by two pathologists. The collected data were evaluated statistically. The research hypothesis was verified with regard to histopathological changes by the chi-squared test (significant at *P* ≤ 0.05). Computations were performed using STATISTICA 10.0 software (StatSoft Inc. 2017).

The morphometric indices for villus height were measured from the tip of the villi to the top of the crypt, the crypt depth was measured from the base of the villi to the submucosa, the villus width was measured in the middle of the villi, and the muscular wall was measured with an average of 15–20 measures per tissue. Each section was imaged using a Panoramic Scanner MIDI (3DHISTECH, Hungary). The measurement data of the alimentary system were prepared using Panoramic Viewer software (3DHISTECH, Hungary). Two-way ANOVA was performed (the effects of age and EM addition were analysed). The statistical analysis included the determination of the characteristics of the analysed traits (the arithmetic mean and standard error of the mean—SEM) and the significance of differences in mean values between age and EM addition groups by Duncan’s D test (significance was set at *P* ≤ 0.05). The research hypothesis was verified with regard to morphometry by the chi-squared test (significant at *P* ≤ 0.05). Computations were performed using STATISTICA 10.0 software (StatSoft Inc. 2017).

## Figures and Tables

**Figure 1 molecules-26-02786-f001:**
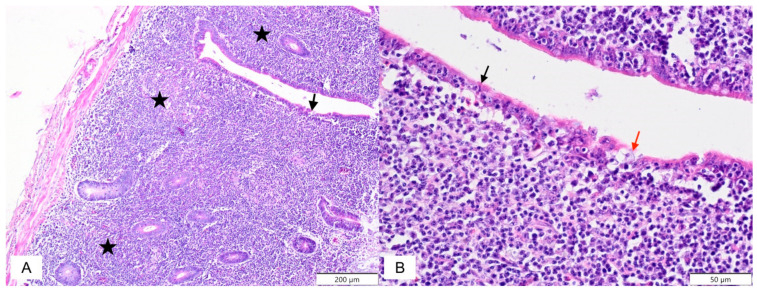
(**A**) Japanese quail. Jejunum. CP group. 3rd day of experiment. Diffuse infiltration of lymphoid cells (asterisk); necrosis of enterocytes (black arrow), (**B**) Higher magnification of Figure 1A, necrosis of enterocytes as indicated by nuclear pyknosis with eosinophilic cytoplasm (black arrow) and nuclear pyknosis with lose of adherence to basement membranes (red arrow). Haematoxylin and eosin staining.

**Figure 2 molecules-26-02786-f002:**
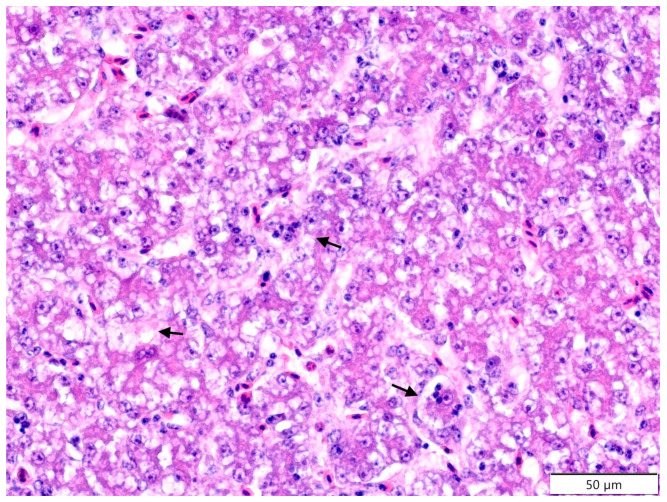
Japanese quail. Liver. CP group. 12th day of experiment. Fatty degeneration of the hepatocytes (black arrows). Detachment of the hepatocytes. Haematoxylin and eosin staining.

**Figure 3 molecules-26-02786-f003:**
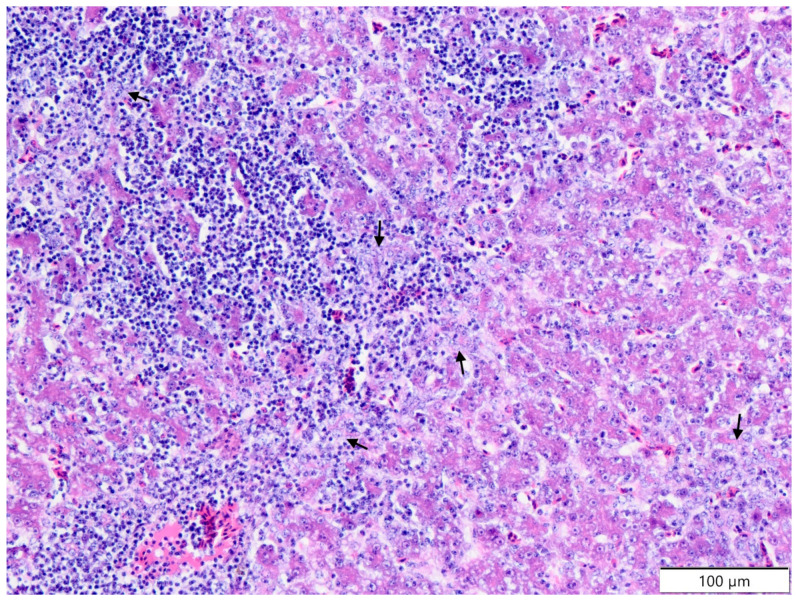
Japanese quail. Liver. CP group. 12th day of experiment. Infiltration of lymphoid cells; proliferation of bile ductules (black arrows). Haematoxylin and eosin staining.

**Figure 4 molecules-26-02786-f004:**
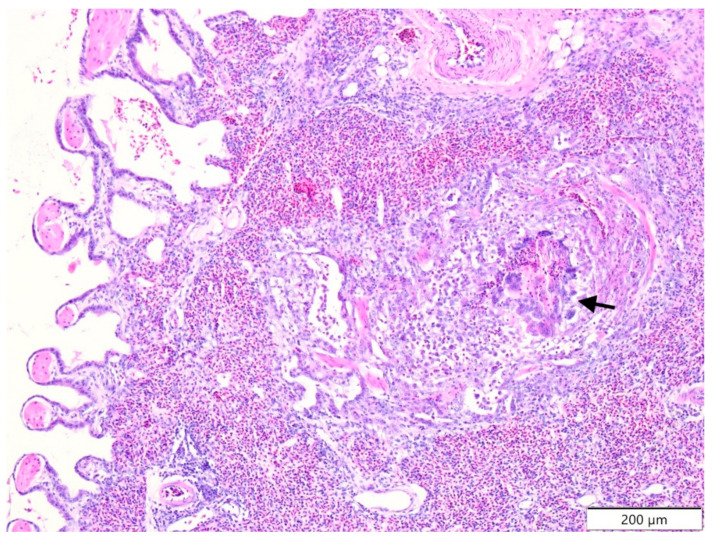
Japanese quail. Lung. CP group. 9th day of experiment. Granulomatous inflammation (black arrow). Congestion of the lung. Haematoxylin and eosin staining.

**Table 1 molecules-26-02786-t001:** Effect of EM supplementation on morphological lesions in the alimentary track in quails challenged with *C. perfringens.*

Morphological Changes *n* = 6	Day of Experiment									
	3		6		9		12		*P*-Value	
	CP	CP + EM	CP	CP + EM	CP	CP + EM	CP	CP + EM	CP	CP + EM
Duodenum									Total number of lesions	
Hyperaemia of the mucosa	0	0	0	0	0	2	0	0	0	2
*P*-value	-		-		0.074		-		0.253	
Focal infiltration of lymphoid cells in the mucosa	3	4	3	3	0	0	0	0	6	7
*P*-value	0.557		0.564		-		-		0.005	
Multifocal infiltration of lymphoid cells in the mucosa	3	2	3	0	3	2	5	1	14	5
*P*-value	0.557		0.023		0.557		0.016		0.043	
Focal infiltration of heterophils cells in the mucosa	0	1	0	0	0	0	0	0	0	1
*P*-value	0.224		-		-		-		0.743	
Irregular surface of the villi	0	0	0	1	0	0	0	0	0	1
*P*-value	-		0.224		-		-		0.743	
Jejunum										
Hyperaemia of the mucosa	2	0	2	0	0	0	0	1	4	1
*P*-value	0.075		0.075		-		0.224		0.122	
Focal infiltration of lymphoid cells in the mucosa	0	2	5	2	0	2	0	2	5	8
*P*-value	0.075		0.502		0.075		0.075		0.028	
Multifocal infiltration of lymphoid cells in the mucosa	3	0	0	1	3	1	0	1	6	3
*P*-value	0.022		0.224		0.213		0.224		0.061	
Diffuse infiltration of lymphoid cells in the mucosa	3	0	1	0	1	0	4	0	9	0
*P*-value	0.022		0.224		0.224		0.006		0.007	
Necrosis of enterocytes	3	0	0	0	0	0	0	0	3	0
*P*-value	0.022		-		-		-		0.049	
Oedema of the villi	2	0	3	0	0	0	0	0	5	0
*P*-value	0.227		0.023		-		-		0.024	
Fusion of the villi	2	0	4	0	0	0	0	0	6	0
*P*-value	0.227		0.006		-		-		0.004	
Irregular surface of the villi	0	2	0	0	0	3	0	0	0	5
*P*-value	0.075		-		0.023		-		0.024	
Goblet cells proliferation	0	0	0	0	2	0	4	0	6	0
*P*-value	-		-		0.075		0.006		0.004	
Ileum										
Hyperaemia of the mucosa	2	0	0	0	0	0	0	0	2	0
*P*-value	0.075		-		-		-		0.253	
Focal infiltration of lymphoid cells in the mucosa	0	0	0	0	2	0	1	0	3	0
*P*-value	-		-		0.075		0.224		0.049	
Multifocal infiltration of lymphoid cells in the mucosa	3	0	0	0	0	0	0	0	3	0
*P*-value	0.022		-		-		-		0.049	
Diffuse infiltration of lymphoid cells in the mucosa	3	0	0	0	0	0	0	0	3	0
*P*-value	0.022		-		-		-		0.049	
Fusion of the villi	0	0	2	0	0	0	0	0	2	0
*P*-value	-		0.075		-		-		0.253	
Irregular surface of the villi	0	0	0	2	0	0	0	0	0	2
*P*-value	-		0.075		-		-		0.253	
Fibromuscular dysplasia in artery	0	1	0	0	0	0	0	0	0	1
*P*-value	0.224		-		-		-		0.743	
Cecum										
Hyperaemia of the mucosa	0	0	1	0	0	0	0	0	1	0
*P*-value	-		0.224		-		-		0.743	

Number in the table represent number of birds with those lesions (*n* = 6). CP—*Clostridium perfringens* infected group; CP + EM—*Clostridium perfringens* infected group with supplementation of effective microorganisms.

**Table 2 molecules-26-02786-t002:** Effect of EM supplementation on morphological lesions in the liver and lungs in quails challenged with *C. perfringens*.

	Day of Experiment									
Morphological Changes *n* = 6	3		6		9		12		*P*-Value	
	CP	CP + EM	CP	CP + EM	CP	CP + EM	CP	CP + EM	CP	CP + EM
Liver									Total number of lesions	
Congestion	0	0	0	0	0	0	1	0	1	0
*P*-value	-		-		-		0.224		0.743	
Parenchymatous degeneration of the hepatocytes	1	2	4	1	4	3	5	3	14	9
*P*-value	0.502		0.071		0.557		0.213		0.153	
Glycogenic degeneration the hepatocytes	0	1	0	0	0	0	0	0	0	1
*P*-value	0.224		-		-		-		0.743	
Hydropic degeneration of the hepatocytes	1	0	4	0	2	3	5	0	12	3
*P*-value	0.224		0.006		0.557		0.024		0.016	
Fatty degeneration of the hepatocytes	1	1	1	3	4	3	5	1	11	8
*P*-value	1.000		0.213		0.557		0.016		0.069	
Focal necrosis of the hepatocytes	1	0	3	0	2	0	4	0	10	0
*P*-value	0.224		0.022		0.075		0.006		0.005	
Necrosis of the bile ductules	0	0	0	0	2	0	3	0	5	0
*P*-value	-		-		0.075		0.022		0.024	
Infiltration of lymphoid cells around bile ductules	1	0	0	0	0	0	3	2	4	2
*P*-value	0.224		-		-		0.557		0.058	
Proliferation of the bile ductules	0	0	0	0	0	0	3	0	3	0
*P*-value	-		-		-		0.022		0.049	
Interstitial infiltration of lymphoid cells	1	0	0	0	0	0	3	2	4	2
*P*-value	0.224		-		-		0.557		0.058	
Lungs										
Congestion	0	1	0	2	3	2	4	2	7	7
*P*-value	0.224		0.075		0.557		0.224		0.069	
Cartilaginous nodules	0	1	1	0	0	0	0	0	1	1
*P*-value	0.224		0.224		-		-		0.562	
Fibromuscular dysplasia	0	0	0	0	0	1	0	0	0	1
*P*-value	-		-		0.224		-		0.743	
Congestion of *lamina propria* in bronchi	0	0	0	1	1	0	0	0	1	1
*P*-value	-		0.224		0.224		-		0.562	
Infiltration of lymphoid cells in parabronchi/lungs	4	1	2	2	3	0	2	0	11	3
*P*-value	0.071		1.000		0.022		0.075		0.050	
Granulomatous inflammation	0	0	0	0	2	0	0	0	2	0
*P*-value	-		-		0.075		-		0.253	

Number in the table represent number of birds with those lesions (*n* = 6). CP—*Clostridium perfringens* infected group; CP + EM—*Clostridium perfringens* infected group with supplementation of effective microorganisms.

**Table 3 molecules-26-02786-t003:** Effect of EM supplementation on morphological lesions in the heart muscle and kidneys in quails challenged with *C. perfringens*.

Morphological Changes *n* = 6	Day of Experiment									
	3		6		9		12		*P*-Value	
	CP	CP + EM	CP	CP + EM	CP	CP + EM	CP	CP + EM	CP	CP + EM
Heart muscle									Total number of lesions	
Congestion of the *miocardium*	1	2	2	1	2	0	5	2	10	5
*P*-value	0.502		0.502		0.075		0.071		0.076	
Degeneration of the cardiomyocytes with vacuolization	2	0	5	3	5	2	5	2	17	7
*P*-value	0.075		0.213		0.071		0.071		0.009	
Eosinophilic sarcoplasm with pyknotic nuclei	1	2	3	2	2	2	2	2	8	8
*P*-value	0.502		0.557		1.000		1.000		0.980	
Atherosclerosis of arteries	1	0	2	1	3	0	0	1	6	2
*P*-value	0.224		0.502		0.022		0.224		0.135	
Heterophilic epicarditis	0	0	0	0	2	0	0	0	2	0
*P*-value	-		-		0.075		-		0.253	
Kidneys										
Congestion	2	3	4	2	3	2	5	2	14	9
*P*-value	0.557		0.244		0.557		0.071		0.514	
Congestion of glomeruli	0	0	2	3	1	1	3	0	6	4
*P*-value	-		0.557		1.000		0.023		0.059	
Degeneration of epithelium cells in proximal tubules	1	2	2	2	3	2	4	1	10	7
*P*-value	0.502		1.000		0.557		0.071		0.651	
Calcium deposits	0	1	0	0	0	0	3	0	3	1
*P*-value	0.224		-		-		0.022		0.055	
Membranous glomerulopathy	0	0	0	0	0	0	1	0	1	0
*P*-value	-		-		-		0.224		0.743	
Infiltration of lymphoid cells around tubules	0	0	0	1	0	0	0	0	0	1
*P*-value	-		0.224		-		-		0.743	
Focal necrosis of epithelial cells in proximal tubules	0	0	0	1	1	0	3	0	4	1
*P*-value	-		0.224		0.224		0.023		0.050	

**Table 4 molecules-26-02786-t004:** Morphometrical analysis of the alimentary system on different days of the experiment.

	Group	Day of Experiment				*P*-Value		
		3	6	9	12	Age	Group	A × G
Duodenum								
Villus height	CP	*1159.99 ^A^	1259.91 ^B^	1065.77 ^C^	1091.99 ^A C^	0.000	0.593	0.039
SEM 15.20	CP + EM	1065.92 ^B^	1238.69 ^A^	1163.76 ^A B^	1059.82 ^B^			
Villus width	CP	192.13	190.88	189.87	201.23	0.057	0.686	0.227
SEM 4.34	CP + EM	177.69 ^B^	177.38 ^B^	214.97 ^A^	217.55 ^A^			
Crypt depth	CP	172.42	173.69	161.56	179.95	0.003	0.668	0.189
SEM 2.98	CP + EM	180.81 ^A^	155.12 ^B^	153.53 ^B^	189.17 ^A^			
Muscular layer thickness	CP	*84.54	*77.68	77.69	78.96	0.000	0.423	0.000
SEM 1.69	CP + EM	64.68 ^A^	66.35 ^A^	85.00 ^B^	*96.16 ^C^			
Jejunum								
Villus height	CP	523.74	570.75	587.73	493.46	0.221	0.004	0.777
SEM 22.23	CP + EM	589.12	678.33	753.05	662.46			
Villus width	CP	151.52	128.74	146.88	150.43	0.000	0.593	0.039
SEM 2.58	CP + EM	124.60	139.89	137.18	142.92			
Crypt depth	CP	126.05	117.52	116.84	127.93	0.217	0.043	0.793
SEM 2.05	CP + EM	110.92	110.48	112.21	121.51			
Muscular layer thickness	CP	51.57 ^A^	57.52	59.68	62.39 ^B^	0.009	0.358	0.523
SEM 1.09	CP + EM	50.60 ^A^	55.73	61.54 ^B^	55.86			
Ileum								
Villus height	CP	*485.28 ^A^	391.15 ^B^	400.59 ^B^	357.13 ^B^	0.030	0.517	0.000
SEM 8.47	CP + EM	380.38 ^A^	375.46 ^A^	426.95 ^A^	*481.46 ^B^			
Villus width	CP	128.19 ^A^	142.06	156.14 ^B^	155.27 ^B^	0.054	0.055	0.212
SEM 2.71	CP + EM	*136.95	127.88	138.63	138.97			
Crypt depth	CP	*112.30	*103.88	96.38	*113.26	0.001	0.013	0.096
SEM 2.46	CP + EM	84.50 ^A B^	80.38 ^A^	104.76 ^C^	100.02 ^B C^			
Muscular layer thickness	CP	*72.72 ^A^	55.92 ^B^	79.24 ^A^	61.50 ^B^	0.000	0.001	0.002
SEM 1.83	CP + EM	47.61 ^A^	54.28 ^A B^	72.78 ^C^	60.66 ^B^			

* Indicates the value within a column with significant differences (*P* > 0.05). ^A–C^ The different superscript letters within rows represent significant differences (*P* > 0.05). CP—*Clostridium perfringens*-infected group; CP + EM—*Clostridium perfringens*-infected group supplemented with effective microorganisms.

**Table 5 molecules-26-02786-t005:** Effect of EM supplementation on blood enzymes, albumin, total protein, and cholesterol level in quails challenged with *C. perfringens*.

	Group	Day of Experiment				*P*-Value		
		3	6	9	12	Age	Group	A × G
Cholesterol (mmol/L)	CP	4.23 ^A^	4.59 ^A^	4.78 ^A^	*6.76 ^B^	0.004	0.890	0.989
SEM 0.26	CP + EM	4.54	4.51	4.79	3.33			
ALT (U/L)	CP	5.53	8.00	3.00	3.00	0.114	0.498	0.270
SEM 0.49	CP + EM	6.00	5.67	5.00	5.33			
AST (U/L)	CP	109.50	137.50	123.50	89.00	0.062	0.278	0.221
SEM 5.25	CP + EM	119.00	94.00	121.00	85.00			
Albumin (g/L)	CP	14.20	16.35	15.70	13.63	0.129	0.047	0.471
SEM 0.76	CP + EM	7.67	14.73	13.23	12.73			
Total protein (g/L)	CP	35.40 ^A^	36.90 ^A^	35.85 ^A^	*65.20 ^B^	0.000	0.000	0.000
SEM 2.33	CP + EM	26.63	35.90	36.00	31.50			
LDH (U/L)	CP	950.50 ^A^	1037.00 ^A^	481.00 ^B^	*1070.00 ^A^	0.003	0.000	0.173
SEM 65.18	CP + EM	462.67 ^A^	802.33 ^B^	358.00 ^A^	490.00 ^A^			

* Indicates the value within a column with significant differences (*P* > 0.05). ^A–C^ The different superscript letters within rows represent significant differences (*P* > 0.05). CP—*Clostridium perfringens*-infected group: CP + EM—*Clostridium perfringens*-infected group supplemented with effective microorganisms. Aspartate aminotransferase (AST) and alanine aminotransferase (ALT).

**Table 6 molecules-26-02786-t006:** Chemical composition of the diets.

Item	Day 0 to 7 ^1^ (Starter)	Day 8 to 28 ^1^ (Grower I)	Day 29 to 42 ^1^ (Grower II)	From Day 42 ^1^ (for Laying Birds)
Chemical composition %				
Crude protein	26.0	23.5	20.0	21.0
Gross fibre	3.0	3.1	3.4	3.8
Lysine	1.50	1.25	1.00	1.15
Methionine	0.69	0.53	0.44	0.45
Methionine + Cysteine	1.10	0.93	0.80	0.81
Ca	1.00	0.88	0.95	3.2
Na	0.17	0.16	0.15	0.16
Cl	0.16	0.15	0.14	0.15
Total P	0.50	0.40	0.40	0.55
ME (kcal/kg)	2975	2900	2800	2800
The vitamin and mineral content				
Vitamin A (IU/kg)	13,200			
Vitamin D3 (IU/kg)	3120			
Vitamin E (IU/kg)	68			
Vitamin K3 (mg/kg)	4.80			
Vitamin B1 (mg/kg)	2.2			
Vitamin B2 (mg/kg)	7.2			
Vitamin B6 (mg/kg)	5.0			
Vitamin B12 (mg/kg)	0.01			
Biotin (mg/kg)	0.03			
Cu (mg/kg)	8			
Fe (mg/kg)	116			
Mn (mg/kg)	80			
Zn (mg/kg)	100			
I (mg/kg)	0.80			
Se (mg/kg)	0.20			

^1^ Ingredients: Corn (24.0%), wheat (38.0%), soybean meal (30.0%), calcium carbonate (5.0%), monocalcium phosphate (1.7%), NaCl (0.3%), and premix (1.0%).

## Data Availability

Data is contained within the article.

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
