# Peer review of "Effective Microorganisms (EM) Improve Internal Organ Morphology, Intestinal Morphometry and Serum Biochemical Activity in Japanese Quails under Clostridium perfringens Challenge"

_molecules, 2021, doi:10.3390/molecules26092786_

Round 1

Reviewer 1 Report

The work presented in this manuscript deserves due consideration.  The experimental part appears to be properly designed and has been executed with due diligence.  Overall, the manuscript presents good quality work. 

On the whole, the manuscript reads well, although there are some problems with sentence structure and grammar.  However, in my opinion as a reader, this manuscript appears excessively drawn-out, and therefore the key findings appear somewhat diluted.   

I find a description of data in Tables somewhat incomplete, which may be confusing for some readers. Please indicate clearly that data for morphological changes for each category refers to the number of birds affected within each treatment group and sampling period.

On the technical side, there are some problems in regards to histopathological lesions data presentation.  Some images appear out of focus and lack proper contrast.  Histological features of tissues stained with haematoxylin and eosin do not show properly on the presented images. In these instances, I would suggest to tone down the amount and hue of blue and increase pink.   

While documentation of diffuse infiltration of inflammatory cells shown on histopathological images of tissues would be acceptable, at the microscope magnification as presented, it is impossible to appreciate features of necrosis shown on images with black arrows. The exaggerated size of arrows distracts attention from the features it supposed to show which are disproportionally small.   

In these instances, I would recommend including higher magnification inserts that would clearly show characteristic features of cell necrosis such as cytoplasmic eosinophilia, chromatin condensation and clamping, and karyorrhexis.  

Al things considered, overall, this manuscript appears sound and presents findings that would be suitable for publication.   

Author Response

Review 1

The work presented in this manuscript deserves due consideration.  The experimental part appears to be properly designed and has been executed with due diligence.  Overall, the manuscript presents good quality work. 

Reply: Thank you for overall mark.

On the whole, the manuscript reads well, although there are some problems with sentence structure and grammar.  However, in my opinion as a reader, this manuscript appears excessively drawn-out, and therefore the key findings appear somewhat diluted.   

Reply: Our paper was edited for proper English language, grammar, punctuation, spelling, and overall style by one or more of the highly qualified native English speaking editors at American Journal of Experts. (Order ID: MMS2LYNZ). I put the certificate at the end of message. I know the paper is a little drawn-out, but I did not find any way to present all obtained data. Microscopic evaluation always bring plenty of information and tables are required for that. Discussion part also must include comparison of obtained data with other researchers.   

I find a description of data in Tables somewhat incomplete, which may be confusing for some readers. Please indicate clearly that data for morphological changes for each category refers to the number of birds affected within each treatment group and sampling period.

Reply 1: Thank you for suggestion. Under each table we give description: “Number in the table represent number of birds with those lesions (n=6).”

On the technical side, there are some problems in regards to histopathological lesions data presentation.  Some images appear out of focus and lack proper contrast.  Histological features of tissues stained with haematoxylin and eosin do not show properly on the presented images. In these instances, I would suggest to tone down the amount and hue of blue and increase pink.   

Reply: Sorry for that. During putting figures in submitting website quality of the pictures has been lowered. We prepared high quality pictures but I see what is the matter of problem. Now we present higher magnification to show lesions more pronounced.

While documentation of diffuse infiltration of inflammatory cells shown on histopathological images of tissues would be acceptable, at the microscope magnification as presented, it is impossible to appreciate features of necrosis shown on images with black arrows. The exaggerated size of arrows distracts attention from the features it supposed to show which are disproportionally small.   

Reply: Now figure was added with higher magnification

In these instances, I would recommend including higher magnification inserts that would clearly show characteristic features of cell necrosis such as cytoplasmic eosinophilia, chromatin condensation and clamping, and karyorrhexis.  

Reply: Now figure was added with higher magnification

Al things considered, overall, this manuscript appears sound and presents findings that would be suitable for publication.   

Reply: Thank you one more time.

Reviewer 2 Report

Well written and comprehensive approach to studying effective microorganisms (EM) as a potential buffer against development of clostridiosis in quail.

Major recommendations for improvement:

Would recommend adding detail in the introduction about the results/outcomes (beneficial/detrimental?) of the prior probiotic/EM research described, rather than only describing what aspects of production were studied in that research.  

Would recommend simplifying tables 1-3 to reduce clutter, horizontal lines, and superfluous text.  Would recommend higher magnification figures to demonstrate cellular/nuclear detail of necrosis.  I fully believe pathologists' findings/expertise, but readers need to be able to see detail for themselves in the figures to corroborate and have faith in your findings.

Would recommend reorganizing results/analysis and discussion to focus on liver lesions then intestinal changes, then minor findings, since liver lesions most closely match clinical expression of quail Clostridiosis.  Agree that enteric findings likely indicate subclinical disease expression, but are vague/ less specific in comparison to the liver lesions presented. 

Specific comments for clarification line by line:

Line 34: "Clostridial organisms are..." then continue with gram-positive bacteria, widely distributed. )need to distinguish from all Gram+ bacteria which is how sentence reads currently."

Line 39: "Predisposing factors also result... bacterial toxins"  Add "also" and "bacterial" to show what is occurring from predisposing factor and what is resulting from the bacteria.  As written, "toxins" could be interpreted as predisposing factors from some other cause.

45. May want to define "probiotics" and "prebiotics".

46. What did Timmerman study find?  otherwise, this sentence does not contribute any relevant information. Rather than telling about all the types of challenges each research group perfromed 52-54 and in lines 67-78, describe what was found from their experiments.  Was there a beneficial or detrimental effect?

49. Rephrase "Researchers have started to challenge the supplementation of probiotic organisms with experimental infections and C. perfringens infection, causing enteritis."  otherwise it sounds like the probiotic models cause the enteritis rather than are being tested for efficacy in reducing enteritis in C. perfringens trials.

80. improving > proving.  Has not yet been tested/proved, so cannot be improved upon.

93. remind reader of CP group meaning, since Clostridium perfringens was written out in full in intro paragraphs, or alternately introduce the abbreviation CP in intro.

The complexity and "busyness" of tables 1-3 are overwhelming to reader. Table formatting should be simplified if possible.  For instance, could descriptive headers be reduced to single line of text ("Hepatocyte degeneration" "Mucosal lymphoid infiltration-focal").  If possible, is there a way to reduce the number of horizontal lines in this table?  Perhaps through use of shading for P value entry lines.  Why group Liver and Lungs together?  Could also simplify by splitting apart for each system. 

Figure 1.  Agree on infiltration by lymphocyte populations, but image is too low power to accurately evaluate enterocyte necrosis.  There is also minor tissue sectioning artifact (chatter) which could explain some separation of mucosal cells from underlying connective tissue by the arrow.  Could authors provide higher magnification showing more cellular/nuclear detail to prove enterocyte necrosis?  Recommend stating level of magnification and animal species in figure legends for all microscopic images as well.

Figure 2/3: Again, agree on lymphoid infiltrates within parenchyma and adjacent to portal vasculature, but the image is at too low magnification to confirm nuclear/cytoplasmic detail for diagnosis of "necrosis."  Please provide higher magnification of these images to show necrosis in greater cellular/nuclear detail.  I can just make out some potential cytoplasmic and/or nuclear condensation, but would need higher magnification to confirm this is true change/lesion and not artifact.

Fig. 4- Need higher magnification of image (or an inset of part of the image at higher magnification) to show the biliary proliferation.  Not easily apparent at this level of magnification.  For this image, I would recommend keeping a low power of the whole field showing the significant lymphoid infiltration, and then add an inset or 2nd panel photo of the biliary proliferation.

Fig 5. Would recommend describing the rest of the hypercellularity throughout- is the rest of this all marked pulmonary congestion?  Appears so, but difficult to tell at this magnification.

Table 4 and 5 are much more simply arranges and seem less cluttered than earlier tables.

81.  Change "data > "results"  

198. Meaning of this statement is unclear: "thus, protecting the anti-inflammatory activity of EM is visible".  Consider rephrasing.  Did you mean there is discernable anti-inflammatory protection by EM?

201- add "individual or small clusters" of necrotic enterocytes to distinguish from the more widespread necrosis observed in fulminant NE

235- "In our opinion, the lesions observed in the liver are more severe than those observed in the gut. Necrotic lesions and proliferation of bile ductules are more typical for C. perfringens infection than inflammatory reactions in alimentary tracks. Clostridium perfringens in quails cause more typical lesions for C. perfringens in the liver than the alimentary track."  I would agree.  Perhaps it would be worthwhile to reorganize the results/discussion to focus around the liver lesions which more typify CP in quail and step down to the enteric lesions and other findings.  Start with most consistent with disease expression and work outward.  From the figures, I would argue that your liver lesions are more typical and important than the intestinal, unless more evidence of enterocyte necrosis is evident at higher magnification.

294- at what stocking density or what cage area?  What lighting regimen was used regarding feed/water intake?

311 or  314- what quantity/dose of CP was gavaged?

315- on blood sampling- what anatomic location for sampling? what quantity of sample? using what needle gauge?

298- what was the type of water/water source for the EM mixture?

Author Response

Review 2

Well written and comprehensive approach to studying effective microorganisms (EM) as a potential buffer against development of clostridiosis in quail.

Major recommendations for improvement:

Would recommend adding detail in the introduction about the results/outcomes (beneficial/detrimental?) of the prior probiotic/EM research described, rather than only describing what aspects of production were studied in that research.  

Reply 1: In introduction we give details about results with probiotic and EM addition.  

Would recommend simplifying tables 1-3 to reduce clutter, horizontal lines, and superfluous text.  Would recommend higher magnification figures to demonstrate cellular/nuclear detail of necrosis.  I fully believe pathologists' findings/expertise, but readers need to be able to see detail for themselves in the figures to corroborate and have faith in your findings.

Reply 2: I reduce horizontal lines and right now for me tables are unreadable. But it is Editor decision, how to create the tables. All figure now demonstrate pathological details. 

Would recommend reorganizing results/analysis and discussion to focus on liver lesions then intestinal changes, then minor findings, since liver lesions most closely match clinical expression of quail Clostridiosis.  Agree that enteric findings likely indicate subclinical disease expression, but are vague/ less specific in comparison to the liver lesions presented. 

Reply: That is true that liver lesion are more typical for C.perfringens infectious, but we diagnosed more lesions in alimentary system. Fearing to avoid chaos in paper I recommend to leave it in present form.   

Specific comments for clarification line by line:

Line 34: "Clostridial organisms are..." then continue with gram-positive bacteria, widely distributed. )need to distinguish from all Gram+ bacteria which is how sentence reads currently."

Reply 3: Yes. You are absolutely right. Sorry for that.

Line 39: "Predisposing factors also result... bacterial toxins"  Add "also" and "bacterial" to show what is occurring from predisposing factor and what is resulting from the bacteria.  As written, "toxins" could be interpreted as predisposing factors from some other cause.

Reply 4: Thank you for that. Now this sentence is understandable.

  1. May want to define "probiotics" and "prebiotics".

Reply: I think that readers already knows what pro, pre and synbiotic are. The article is already long enough.

  1. What did Timmerman study find?  otherwise, this sentence does not contribute any relevant information. Rather than telling about all the types of challenges each research group perfromed 52-54 and in lines 67-78, describe what was found from their experiments.  Was there a beneficial or detrimental effect?

Reply 1:  I rewrite this chapter and add information about beneficial effects of the supplementation.

  1. Rephrase "Researchers have started to challenge the supplementation of probiotic organisms with experimental infections and C. perfringens infection, causing enteritis."  otherwise it sounds like the probiotic models cause the enteritis rather than are being tested for efficacy in reducing enteritis in C. perfringens trials.

Reply 5: The sentence was delete. Now was added - “Few studies have been conducted evaluating the implication of probiotic organisms supplementation during C. perfringens experimental  infection”

  1. improving > proving.  Has not yet been tested/proved, so cannot be improved upon.

Reply 6: Yes, you are right. I made correction.

  1. remind reader of CP group meaning, since Clostridium perfringens was written out in full in intro paragraphs, or alternately introduce the abbreviation CP in intro.

Reply: I introduce the abbreviation in intro. Thank you.

The complexity and "busyness" of tables 1-3 are overwhelming to reader. Table formatting should be simplified if possible.  For instance, could descriptive headers be reduced to single line of text ("Hepatocyte degeneration" "Mucosal lymphoid infiltration-focal").  If possible, is there a way to reduce the number of horizontal lines in this table?  Perhaps through use of shading for P value entry lines.  Why group Liver and Lungs together?  Could also simplify by splitting apart for each system. 

Reply: As you suggested I deleted all horizontal lines and right now for me tables are unreadable. But it is Editor decision, how to create the tables. I am sure that tables will look different in final version on website. The liver and lungs are together in table because I have tried to create similar size two tables. If I put liver and kidney together, other will ask why I did that…. 

Figure 1.  Agree on infiltration by lymphocyte populations, but image is too low power to accurately evaluate enterocyte necrosis.  There is also minor tissue sectioning artifact (chatter) which could explain some separation of mucosal cells from underlying connective tissue by the arrow.  Could authors provide higher magnification showing more cellular/nuclear detail to prove enterocyte necrosis?  Recommend stating level of magnification and animal species in figure legends for all microscopic images as well.

Reply: Now figure was added with higher magnification showing clusters of enterocytes necrosis. Animal species was added in all figures. Nowadays scale bar always is burned into figure, the is no need to give magnification number. 

Figure 2/3: Again, agree on lymphoid infiltrates within parenchyma and adjacent to portal vasculature, but the image is at too low magnification to confirm nuclear/cytoplasmic detail for diagnosis of "necrosis."  Please provide higher magnification of these images to show necrosis in greater cellular/nuclear detail.  I can just make out some potential cytoplasmic and/or nuclear condensation, but would need higher magnification to confirm this is true change/lesion and not artifact.

Reply: Now figure was added with higher magnification

Fig. 4- Need higher magnification of image (or an inset of part of the image at higher magnification) to show the biliary proliferation.  Not easily apparent at this level of magnification.  For this image, I would recommend keeping a low power of the whole field showing the significant lymphoid infiltration, and then add an inset or 2nd panel photo of the biliary proliferation.

Reply: Now figure was added with higher magnification

Fig 5. Would recommend describing the rest of the hypercellularity throughout- is the rest of this all marked pulmonary congestion?  Appears so, but difficult to tell at this magnification.

Reply: I added “congestion of the lung” into text.

Table 4 and 5 are much more simply arranges and seem less cluttered than earlier tables.

Reply: because we diagnosed there less lesions.

  1. Change "data > "results"  

Reply 7: I put corrections.

  1. Meaning of this statement is unclear: "thus, protecting the anti-inflammatory activity of EM is visible".  Consider rephrasing.  Did you mean there is discernable anti-inflammatory protection by EM?

Reply 8: Yes. I mean anti-inflammatory protection by EM. I change it.

201- add "individual or small clusters" of necrotic enterocytes to distinguish from the more widespread necrosis observed in fulminant NE

Reply 9: thank you. Great idea.

235- "In our opinion, the lesions observed in the liver are more severe than those observed in the gut. Necrotic lesions and proliferation of bile ductules are more typical for C. perfringens infection than inflammatory reactions in alimentary tracks. Clostridium perfringens in quails cause more typical lesions for C. perfringens in the liver than the alimentary track."  I would agree.  Perhaps it would be worthwhile to reorganize the results/discussion to focus around the liver lesions which more typify CP in quail and step down to the enteric lesions and other findings.  Start with most consistent with disease expression and work outward.  From the figures, I would argue that your liver lesions are more typical and important than the intestinal, unless more evidence of enterocyte necrosis is evident at higher magnification.

Reply: That is true that liver lesion are more typical for C.perfringens infectious, but we diagnosed more lesions in alimentary system and they are more severe. Fearing to avoid chaos in paper I recommend to leave it in present form.   

294- at what stocking density or what cage area?  What lighting regimen was used regarding feed/water intake?

Reply 10: Density in cages (0,040 m2/bird). Lighting period for laying quails - 16 hours. - All info added to the text.

311 or  314- what quantity/dose of CP was gavaged?

Reply 11: 1.0 ml/bird . added.

315- on blood sampling- what anatomic location for sampling? what quantity of sample? using what needle gauge?

Reply 12: Blood samples were taken from brachial/wing vein. When was problems with collecting, we used jugular vein. Quantity of sample - 2.0 - 3.0 ml. needle 33G (0,2mm) - 30 G (0.3mm)  . Anatomic location was added to the text.

298- what was the type of water/water source for the EM mixture?

Reply 13: normal water/tap water. - according to the EM manufacturer’s specification

Round 2

Reviewer 2 Report

The revised version of the manuscript is much improved in both clarity and completeness of background information and additional detail presented in experimental methods.  Revisions significantly enhanced my understanding of prior work in the field and the current research study.

Would still recommend further minor revision to Figures, though the new images were clearer.  Note: In my version of the revised manuscript, no scale bars were evident in Figs 1-3, so would recommend adding magnification to figure legends or placing scale bars in images as needed. 

In Figure 1, I would recommend making a panel using the original (1a, low mag) and the new figure (1b) side by side.  The original figure clearly showed the significant lymphoid infiltration around jejunal crypts (i.e. showed tissue architecture better) and the new figure shows the close up of the epithelium.  Together, these are a powerful pair.  I think the left-most arrow in the new figure 1 needs to be moved one cell to the right over the brightly eosinophilic enterocyte with a condensed (pyknotic) nucleus and no discernable brush border.  Where presently placed, I cannot tell if that is showing necrotic debris or just infiltrating inflammatory cells.  Looks like brush border is intact where currently placed.  Would also recommend adding a description in the figure legend of what cellular/nuclear features indicated necrosis ("as indicated by...") to the pathologists.

Figure 2: Hepatic necrosis is still not evident where arrows are currently placed or at current magnification.  Arrows appear to be currently placed at edge of sinusoids or intercellular space of detaching (degenerate, but viable) hepatocytes.  The majority of cell nuclei in the image appear to be intact and open/viable (showing little to no pyknosis/karyorrhexis/karyolysis which would indicate necrosis) with little change to cytoplasmic staining (no significant hypereosinophilia or loss of differential staining which would indicate necrosis).  The hepatocyte by the lower right arrow appears to have some inflammatory (lymphocytic) infiltrates around it and may be in the process of detaching from other cells, but the nuclus and cytoplasm still appear viable at present.  Would recommend replacing with image of more evident cellular necrosis or describing cellular/nuclear features of necrosis that were used to arrive at diagnosis.  Alternately, I would agree on describing degeneration and detachment of hepatocytes in this image, but would not define as necrosis in figure legend.

Figure 3. While I feel readers would still benefit from higher magnification view of biliary proliferation to be able to better discern biliary epithelium from surrounding hepatocytes, this magnification does show the lesion adequately enough.  The widespread nature of the lymphocytic infiltrates is best viewed at this magnification, so this image will suffice.

I agree that having no horizontal lines in the tables interferes with readability as well, but the original version had too many lines, creating a busyness that interfered with reader flow too.  I leave the final desicion regarding table formating to the discretion of the editor.  Perhaps an intermediate solution- one line above each microscopic characteristic and one below the corresponding P value, rather than adding an extra line between the characteristic and the P value too?

Thank you for your rapid and significant revisions to your manuscript.  This effort has further strengthened your detailed study.  

Author Response

The revised version of the manuscript is much improved in both clarity and completeness of background information and additional detail presented in experimental methods.  Revisions significantly enhanced my understanding of prior work in the field and the current research study.

Reply: Thank you.

Would still recommend further minor revision to Figures, though the new images were clearer.  Note: In my version of the revised manuscript, no scale bars were evident in Figs 1-3, so would recommend adding magnification to figure legends or placing scale bars in images as needed. 

In Figure 1, I would recommend making a panel using the original (1a, low mag) and the new figure (1b) side by side.  The original figure clearly showed the significant lymphoid infiltration around jejunal crypts (i.e. showed tissue architecture better) and the new figure shows the close up of the epithelium.  Together, these are a powerful pair.  I think the left-most arrow in the new figure 1 needs to be moved one cell to the right over the brightly eosinophilic enterocyte with a condensed (pyknotic) nucleus and no discernable brush border.  Where presently placed, I cannot tell if that is showing necrotic debris or just infiltrating inflammatory cells.  Looks like brush border is intact where currently placed.  Would also recommend adding a description in the figure legend of what cellular/nuclear features indicated necrosis ("as indicated by...") to the pathologists.

Reply 1: Thank you for help. I put two figures 1a and 1b. I think figures are more visible.

Figure 2: Hepatic necrosis is still not evident where arrows are currently placed or at current magnification.  Arrows appear to be currently placed at edge of sinusoids or intercellular space of detaching (degenerate, but viable) hepatocytes.  The majority of cell nuclei in the image appear to be intact and open/viable (showing little to no pyknosis/karyorrhexis/karyolysis which would indicate necrosis) with little change to cytoplasmic staining (no significant hypereosinophilia or loss of differential staining which would indicate necrosis).  The hepatocyte by the lower right arrow appears to have some inflammatory (lymphocytic) infiltrates around it and may be in the process of detaching from other cells, but the nuclus and cytoplasm still appear viable at present.  Would recommend replacing with image of more evident cellular necrosis or describing cellular/nuclear features of necrosis that were used to arrive at diagnosis.  Alternately, I would agree on describing degeneration and detachment of hepatocytes in this image, but would not define as necrosis in figure legend.

Reply 2: Thank you. You are right. Maybe details regarding necrosis are not so visible. We change description.

Figure 3. While I feel readers would still benefit from higher magnification view of biliary proliferation to be able to better discern biliary epithelium from surrounding hepatocytes, this magnification does show the lesion adequately enough.  The widespread nature of the lymphocytic infiltrates is best viewed at this magnification, so this image will suffice.

Reply: Thank you.

I agree that having no horizontal lines in the tables interferes with readability as well, but the original version had too many lines, creating a busyness that interfered with reader flow too.  I leave the final desicion regarding table formating to the discretion of the editor.  Perhaps an intermediate solution- one line above each microscopic characteristic and one below the corresponding P value, rather than adding an extra line between the characteristic and the P value too?

Reply: I think that Editor has final decision, how those table should look like. Pleases give me information what shall I do…..

Thank you for your rapid and significant revisions to your manuscript.  This effort has further strengthened your detailed study.